# Analysis of the Clinical and Epidemiological Meaning of Screening Test for SARS-CoV-2: Considerations in the Chronic Kidney Disease Patients during the COVID-19 Pandemic

**DOI:** 10.3390/jcm10051139

**Published:** 2021-03-09

**Authors:** Francesca Martino, Gianpaolo Amici, Stefano Grandesso, Rosella Ferraro Mortellaro, Antonina Lo Cicero, Giacomo Novara

**Affiliations:** 1UO Nephrology, Dialysis, and Transplantation, San Bortolo Hospital, 36100 Vicenza, Italy; 2UO Nephrology and Dialysis, San Daniele del Friuli and Tolmezzo Hospital, ASUFC, 33038 San Daniele del Friuli, Italy; amicig@tin.it (G.A.); rosella.ferraro-mortellaro@asufc.sanita.fvg.it (R.F.M.); antonina.locicero@asufc.sanita.fvg.it (A.L.C.); 3Laboratory Medicine, Dolo-Mirano District, AULSS 3 Serenissima, 30100 Venezia, Italy; stefano.grandesso@aulss3.veneto.it; 4Department of Surgery, Oncology, and Gastroenterology, Urology Clinic University of Padua, 35124 Padua, Italy

**Keywords:** COVID-19, chronic kidney disease, screening text, accuracy

## Abstract

The COronaVIrus Disease 19 (COVID-19) pandemic is an emerging reality in nephrology. In a continuously changing scenario, we need to assess our patients’ additional risk in terms of attending hemodialysis treatments, follow-up peritoneal dialysis, and kidney transplant visits. The prevalence of severe acute respiratory syndrome coronavirus 2 (SARS-CoV-20 infection in the general population plays a pivotal role in estimating the additional COVID-19 risk in chronic kidney disease (CKD) patients. Unfortunately, local prevalence is often obscure, and when we have an estimation, we neglect the number of asymptomatic subjects in the same area and, consequently, the risk of infection in CKD patients. Furthermore, we still have the problem of managing COVID-19 diagnosis and the test’s accuracy. Currently, the gold standard for SARS-CoV-2 detection is a real-time reverse transcription-polymerase chain reaction (rRT-PCR) on respiratory tract samples. rRT-PCR presents some vulnerability related to pre-analytic and analytic problems and could impact strongly on its diagnostic accuracy. Specifically, the operative proceedings to obtain the samples and the different types of diagnostic assay could affect the results of the test. In this scenario, knowing the local prevalence and the local screening test accuracy helps the clinician to perform preventive measures to limit the diffusion of COVID-19 in the CKD population.

## 1. Introduction

COronaVIrus Disease 19 (COVID-19) is a pandemic disease currently present in more of 200 countries globally [1], and it is caused by severe acute respiratory syndrome coronavirus 2 (SARS-CoV-2). COVID-19 can manifest in different ways from asymptomatic form to severe pneumonia and fatal multi-organ failure. The government in Italy and other countries decided to impose lockdown for an extended period to avoid a wide diffusion of COVID-19 and dilute the correlated need for severe cases of hospitalization. We started limiting unnecessary motion and promoting social distance, mask use, and hand cleaning in this context.

Chronic kidney disease (CKD) patients, especially those in stage 5D, have mandatory needs to attend hospital facilities visits or treatments. This condition per se could increase the risk of COVID-19 and should be carefully evaluated by nephrologists considering their local situation.

## 2. COVID-19 Risk in Chronic Kidney Patients

In the general population, the typical symptoms at onset are fever, dry cough, fatigue, and dyspnoea. Still, in some cases, the patients can present headaches, diarrhea, vomiting, abdominal pain, and dizziness [2,3]. Furthermore, physicians have reported other manifestations such as rash, eye abnormalities, and neurological and heart complications [2,3,4]. The clinical manifestations seem worse in elderly patients and those with comorbidities such as diabetes, hypertension, chronic kidney disease, chronic obstructive pulmonary disease (COPD), and chronic heart disease [5]. The presence of underlying kidney disease seems a risk factor for developing severe complications and appears to be associated with a higher mortality rate [6]. Specifically, a metanalysis including 1389 COVID-19 patients showed an odds ratio (OR) as high as 3 to have severe COVID-19 in the patients with previous CKD [7]. Additionally, CKD patients often suffer from hypertension, diabetes, and heart disease, which are consolidated risk factors for the deleterious progression of COVID-19 [5].

Generally, CKD patients present the same symptoms and signs of the general population [8]. On the basis of literature reports, we reported in Table 1 symptoms and laboratory features common in CKD patients with COVID-19 and their meaning in risk analysis for the development of severe complications such as acute respiratory distress syndrome (ARDS) and death.

Furthermore, the same considerations are substantially valid in kidney transplant patients, in whom the most common symptoms of COVID-19 onset were fever and dyspnea, followed by diarrhea and myalgia [10]. Specifically, in this class of patients, the mortality rate seems to be influenced by the age (OR 1.07), the respiratory rate at presentation >20 breaths/min (OR 6.88), and the kidney function evaluated by estimated Glomerular Filtrate Rate (OR 0.96).

The early phase presentation is not specific to COVID-19, making it difficult to recognize the exordia of disease and prevent diffusion and severe complications. Accordingly, nephrology and dialysis units adopted special programs to individualize potential COVID-19 patients. At every dialysis session or nephrology consult before the facility access, healthcare workers provide a simple triage, evaluating the presence of symptoms and detecting the presence of high temperature and lower O_2_ saturation [11,12]. In doubtful cases, patients are tested for SARS-CoV-2 and start quarantine until exclusion or confirmation of COVID-19 diagnosis. In COVID-19 diagnosis, CKD patients should be ideally transferred to a designated hospital or ward for COVID-19 patients if they need hospitalization. The in-hospital patients who require renal replacement therapy should be treated in an isolation room, and their healthcare workers should wear personal protective equipment (such as KF94 or N95 masks, gloves, goggles, or face shield, level D gown) when performing dialysis [13].

All previous procedures try to limit diffusion in the nephrology and dialysis unit. COVID-19 diffusion control shows its weakness in transmitting SARS-CoV-2 by asymptomatic people to fragile subjects in the general population [14] and potentially can affect CKD patients. However, the transmission by asymptomatic people seems to have a doubtful impact on dialysis patients as suggested by a Lombard study. This study showed a similar positive rate in real-time reverse transcription-polymerase chain reaction (rRT-PCR) in the hemodialysis unit where all patients were screened and in units where only symptomatic patients were screened [9]. This phenomenon could have more than an explanation if, on the one hand, hemodialysis patients could be higher susceptible to severe complications in most of the cases.

On the other hand, the rRT-PCR screening test could be less sensitive in asymptomatic patients with lower viral load. In any case, transmission by asymptomatic people seems to be a reasonable problem and limits our ability to prevent COVID-19 diffusion. Therefore, we can only take prophylactic measures such as: educating patients and healthcare workers about the personal protective dispositive (e.g., masks, and gloves) and social distance; preparing appropriate waiting rooms or resting areas; providing surgical masks and hand disinfection before entering the Hemodialysis (HD) unit [15,16,17].

In a recent survey promoted by the Società Italiana Nefrologia (SIN) [18] on 358 centers, the authors reported a prevalence of COVD-19 equal to 3.41%, 1.36%, and 0.87% in hemodialysis, peritoneal dialysis, and kidney transplant patients, respectively. Unfortunately, only 15% of centers performed at least one screening test on all patients. Furthermore, the authors reported a high death rate in CKD patients with SARS-CoV-2 infection: 49% of mortality in peritoneal dialysis patients, 37% of death in hemodialysis patients, and 25% in kidney transplant patients. On the basis of this preliminary report, in Italy, we see that the diffusion of COVID-19 in CKD patients seems higher than in the general population (as reported by the last updating of Istituto Superiore di Sanità (ISS), the rate of patients positive for SARS-CoV-2 was about 0.362% [218, 268/60, 317,000]). Furthermore, the crude mortality rate in CKD patients with COVID-19 is higher than in the general population, estimated in the same period by the last ISS report, at around 13.9% (30, 395/218, 268). The differences between CKD patients and the general population confirm our patients’ fragility in terms of comorbidities and suggest a higher risk in the people who need frequent access to hospital facilities.

Furthermore, the Registry of the Spanish Society of Nephrology [8] confirms the same trend in COVID-19 dialysis and kidney transplant patients with a high rate of mortality (about 23%) and a high need for hospital admission (about 85%).

Finally, in a multicenter Turkish study on 1210 subjects, dialysis need, kidney transplant, and stage III-V CKD severely impacted on the patient prognosis, resulting in a higher rate of severe COVID-19 (25.4%, 21%, and 39.4%, respectively), and increased mortality (16.2%, 11.1%, and 28.4%, respectively) compared to the patients without kidney disease, for whom severe COVID-19 had a rate of about 8% and mortality of around 4% [19].

On the basis of previous considerations, we cannot consider the standard balance between the risk and the benefit enough for every procedure during the COVID-19 pandemic. Nephrologists have to know SARS-CoV-2 screening tests and their ability to predict COVID-19 to take adequate prophylactic measures to benefit each patient while considering the real risk. Specifically, SARS-CoV-2 screening test accuracy should be considered in patients who wait for a kidney transplant for the need to assess the balance between the risk and the benefit of the procedure in little time [20].

## 3. SARS-CoV-2 and Screening Test

### 3.1. SARS-CoV-2 Structure

SARS-CoV-2 is an RNA single-stranded virus belonging to the family of Coronaviridae, which is divided into four subfamilies: alfa, beta, gamma, and delta. SARS-CoV-2 belongs to beta-coronaviruses and shares at least 50% of its genome with other beta-coronaviruses SARS-CoV and Middle East respiratory syndrome coronavirus (MERS-CoV) members. During replication, SARS-CoV-2 produces 16 non-structural proteins and 6-9 structural and accessory proteins, such as spike (S), envelope (E), membrane (M), and nucleocapsid (N) [21]. Each protein is encoded by a corresponding gene, targeted as N, E, S, and RNA-dependent RNA polymerase (RdRp) genes. Figure 1 reports the SARS-CoV-2 structure and RNA sequences.

### 3.2. Screening Test Methods

Currently, the diagnosis of COVID-19 is confirmed by nucleic acid amplification tests (NAAT), such as real-time reverse transcription polymerase chain reaction (rRT-PCR) on the respiratory tract specimens [22].

After the extraction of RNA, rRT-PCR consists of a three-step procedure:-Reverse transcription: a process where the enzyme reverse transcriptase converts RNA into complementary DNA (cDNA), which is suitable for PCR.-Amplification of cDNA target sequences, which requires the presence of a polymerase enzyme and primer. The polymerase amplifies the cDNA sequence, while the primer identifies the specific sequences to amplify.

Detection, involves fluorescently labelling DNA oligonucleotides, which bind the primer and give a fluorescent signal at each amplification cycle. The fluorescence signal increases as more copies of DNA are produced; when the fluorescence arises to a certain threshold, the test is considered positive.

The World Health Organization (WHO) recommended E, N, and RdRp genes as molecular targets for first-line screening, as well as confirmatory tests on a nasopharyngeal or oropharyngeal swab, and on lower respiratory specimens (such as sputum, endotracheal aspirate, and bronchoalveolar lavage). Furthermore, at the website www.who.int/emergencies/diseases/novel-coronavirus-2019/technical-guidance/laboratory-guidance published at 11 September 2020, accessed on 22 January 2021, the WHO provides technical guidance about the over 250 kits disposable on the market. Generally, the commercial kits detect the presence of two or three viral sequences. In the first case, identifying one gene is used as a screening test, while that of the second gene is used as a confirmatory test. In the latter case, a screening test is considered positive only when all genes are detected. Specifically, WHO suggested PCR amplification of the viral E gene as a screening test and amplification of the RdRp region of the orf1b gene as a confirmatory test. Afterwards, on 12 March 2020, the European Centre for Disease Prevention and Control (ECDC) specified no absolute need for a confirmatory test. Specifically, in lower transmission countries, a confirmatory test is always required. In contrast, in the countries with high transmission, a confirmatory test’s performance is only required when the first result is technically not interpretable, or the RT-PCR cycle threshold value is above 35 [23].

### 3.3. Screening Test Accuracy

Despite the gold standard’s endorsement, rRT-PCR is not flawless, and it has shown accuracy problems, which can lead to underrating SARS-CoV-2 infection.

The COVID-19 pandemic is supposed to be a high-prevalence disease with serious consequences for the patients. In this scenario, a screening test should have high sensitivity with a lower false-negative rate. Precisely, a higher rate of false negatives limits the ability of screening tests to recognize the patients with COVID-19 and consequently increases the likelihood to delay the medical care of COVID-19 patients. This aspect is dangerous for CKD patients, who showed high susceptibility to develop serious consequences after SARS-CoV-2 infection. Additionally, a high rate of false negatives is critical during a pandemic because it does not allow one to follow the recommendations to limit the diffusion in the community without extra cost for the health system. Specifically, in CKD patients, a high false-negative rate increases the risk of dissemination in the hospital facility.

Unfortunately, rRT-PCR’s sensitivity rate was estimated to be around 66–80% [24] in a Chinese study of 1014 patients. On the basis of this report, we see that the accuracy of the rRT-PCR test in the diagnosis of COVID-19 seems to be weak and related to different types of issues. In an exciting review by Lippi et al. [25], the authors reported the two kinds of laboratory problems: preanalytical (such as inadequate procedures for collection, handling, transport and storage, collection of inappropriate or unsuitable material, presence of interfering substances) and analytical (such as testing outside the diagnostic window, active viral recombination, use of inadequately validated assays, insufficient harmonization, and instrument malfunctioning). All these procedural matters result in a high risk of a false-negative test.

Furthermore, the commercially available diagnostics kits in rRT-PCR have different characteristics, mainly due to the viral region investigated and the limit of detection (LoD). Noticeably, the higher the LoD, the more risk of false negatives. In Table 2, we present the characteristics of some of the kits mainly used in Italy. Finally, when the clinical picture is strongly suspected for COVID-19 infection, and the swab is repeatedly negative [26,27], and it may be appropriate to carry out a serological investigation to search for IgM and IgG [26].

Finally, we want to highlight how not only the rate of false negatives but also the rate of false positive results negatively influences the management of vulnerable patients. In the first case, as we emphasized in the previous paragraph, there is a high likelihood of contagious between the patients with potentially devastating consequences for the relatively small CKD communities (patients, health workers, and support personnel). Conversely, in the case of false positive tests, there is a waste of resources for the surveillance and the management of standard care, as well as concomitant psycho-physical stress in patients that is highly proven by their basal health conditions.

### 3.4. Specimen Type

Between the preanalytical issues, the most debated argument is the type of specimen. One of the first reports about COVID-19 described a significant difference in the screening test’s sensibility related to the kind of specimen [28]. Specifically, bronchoalveolar lavage fluid seemed to have the best accuracy with a rate of positive equal to 93%, sputum with a rate of 72%, nasal swabs with a rate of 63%, and finally pharyngeal swabs with a rate of 32%.

It seems accepted that the specimen derived from the upper respiratory tract shows its weakness compared with the low respiratory tract, especially during the symptomatic phase. Nasopharyngeal swab seems more suitable than oropharyngeal swab, which appears to have a higher rate of false negatives, as reported by Wang et al. in a comparative study on about 350 patients [29] and by Mohammadi in a recent meta-analysis [30]. Furthermore, saliva (a clear, slightly alkaline liquid secreted into the mouth by the salivary glands and mucous glands) seems to have the same reliability as nasopharyngeal swabs [31,32] and better reliability of oropharyngeal swabs [33], and thus it should be considered as an alternative specimen in the diagnosis of COVID-19 in symptomatic patients. Finally, sputum sampling (fluid coughed up and expectorated from the mouth, composed of saliva and discharges from the respiratory passages such as mucus and phlegm) seems to have higher sensitivity to nasopharyngeal swab. Likely, if other studies support its better sensitivity, in the future, we should consider the sputum as a preferred specimen in diagnosing and monitoring COVID-19 [30].

Furthermore, over the types of specimen, we have to consider the timing of collection. In the week before symptom onset, the viral load could be very low and likely inadequate for the detection by rRT-PCR. Consequently, in this phase of COVID-19, the screening test could have a high likelihood to have false-negative results [34]. As reported, the higher viral loads are detected soon after symptom onset [30] and can persist in throat swabs for more than 30 days [35,36]. The specimens’ types show different accuracy profiling in various phases of COVID-19, likely related to viral load, suggesting the preferable kind of sample and operative conditions, as reported in Table 3.

Consequently, we have to prefer upper respiratory tract specimens in the incubation period, such as nasopharyngeal swabs saliva/sputum collection. While in the symptomatic period, we have to choose the lower respiratory tract specimens (such as bronchoalveolar lavage fluid) in critical patients who require intubation. Finally, during convalescence, we suggest adding fecal/anal swab to the standard nasopharyngeal swab [37].

### 3.5. Statistical Insight on Screening Test

In general, any test has different performances in different settings or applications. In the COVID-19 screening test case, different disease prevalence can lead to surprisingly different interpretations of tests, even with the same value of sensitivity and specificity. Table A1 reports in synthesis the common statistical knowledge and calculations about test performance evaluation. Positive and negative predictivity value has a key role in interpreting a single test result in a clinical setting because it suggests to a physician whether the test results are trustable. In other words, positive predictive value (PPV) offers the probability of having an ill patient when the result of the test is positive, and negative predictive value (NPV) tells of the probability of having healthy patients when the result is negative.

Specifically, in the COVID-19 pandemic, we observed a different prevalence of the disease in the same population, likely related to the seasonal period and the use of adequate prophylactic measures, which have an obvious impact on the interpretation of the screening test for vulnerable patients, such as CKD patients. Unfortunately, it is not always simple to individualize the real prevalence in different areas considering the variable rate of asymptomatic people, the number of screened people, and the frequency of the screening. Despite these considerations, we suggest optimizing the available information such as the number of COVID-19 patients in the local hospitals, as well as the local reports by the authorities to understand the trend in COVID-19 diffusion.

Table A2, Table A3, Table A4 and Table A5 and Figure 2 report some examples relative to the accuracy of the screening tests for COVID-19, which only have an explicative role. The reported examples are extrapolated from the sensibility and specificity described in some studies [24,27] and show how COVID-19 prevalence and the value of sensitivity and specificity can impact the test’s interpretation. Unfortunately, nasopharyngeal swabs’ real sensitivity and specificity are only partially known with a value of sensibility of 66–80% and specificity of 90–95%.

In high prevalence conditions, and optimal sensitivity and specificity of nasopharyngeal swab, a single result gives high values of predictivity, both positive and negative. However, in a low prevalence situation, the same test with the same sensitivity and specificity gives significantly lower positive predictive values. If we consider oropharyngeal swabs, that show low sensitivity and good specificity, resulting in a poor positive predictive value (largely not useful for screening purposes) and good negative predicted value. Specifically, we have worse PPV and better NPV in a low prevalence situation.

## 4. Conclusions

COVID-19 has been shown to be very risky in CKD patients in terms of the development of serious consequences, such as acute respiratory distress syndrome and death. During the COVID-19 pandemic, the screening test for SARS-CoV-2 is considered the gold standard for the diagnosis. Unfortunately, different issues such as the sensibility/specificity of the single test, the period of performing, the type of specimen, and the prevalence of disease could strongly impact on the interpretation of the test and its reliability. In order to reduce the contagious between the patients, nephrologists have to carefully manage the results of the screening test for SARS-CoV-2, considering the suboptimal sensitivity of the test and the relevant likelihood of false-negative results. In this scenario, promoting extensive use of protective measures (such as the personal protective dispositive, social distance, and a limitation of simultaneous access to nephrology facilities) seems a reasonable approach. When possible, considering the local resources, intensifying the number of samples for each patient could be theoretically recommended to overcome the accuracy issue of the screening test. Finally, we suggest considering the anal swab to readmit CKD patients who have SARS-CoV-2 infection to the hospital facilities.

## Figures and Tables

**Figure 1 jcm-10-01139-f001:**
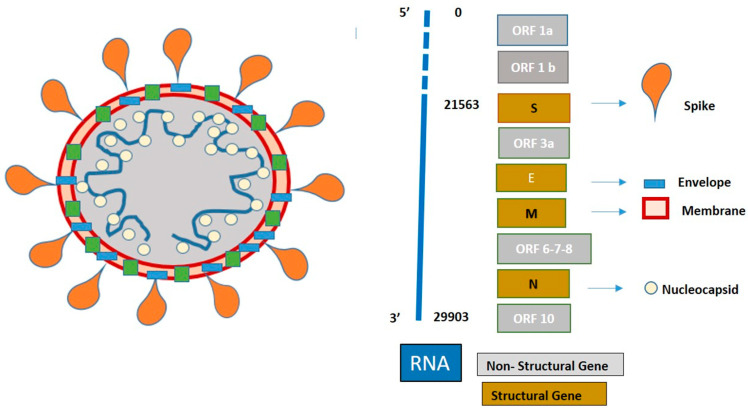
Schematic representation of severe acute respiratory syndrome coronavirus 2 (SARS-CoV-2) structure and genomic.

**Figure 2 jcm-10-01139-f002:**
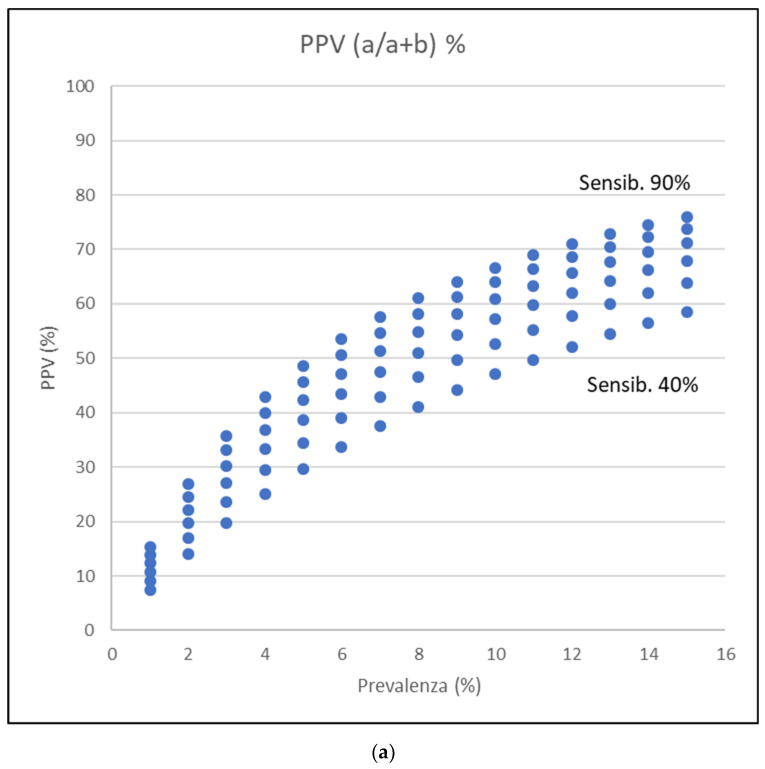
Graphic representation of positive predictive value (PPV) and negative predictive value (NPV) variations with different percentage disease prevalence (*X*-axis, from 1 to 15%), sensibility variation from 40 to 90%, and fixed specificity at 95%. a: true positive, b: false positive, c: false negative, d: true negative. (**a**) PPV values show an increase with increasing disease prevalence. (**b**) NPV values are instead decreasing with disease prevalence increase.

**Table 1 jcm-10-01139-t001:** Significance of symptoms and signs of COronaVIrus Disease 19 (COVID-19) in chronic kidney disease (CKD) patients [9].

Symptoms/Signs	Increased Risk of ARDS	Increased Risk of Death
Cough	At onset ≈	At onset +
Fever	At onset +++	At onset +++
Shortness of breath	At onset +++	At onset ++
Gastrointestinal symptoms nausea vomiting diarrhea	Not significant	Not significant
Pharyngitis	Not significant	Not significant
Shortness of breath	Not significant	Not significant
Myalgia	At onset ++	Not significant
Blood examination	Not significant	Not significant
Lymphocytes decrease	Not significant	Not significant
Platelets decrease	Not significant	Not significant
C-RP increase	>50 mg/L +	>50 mg/L ++
AST/ALT increase	>50 U/L +	Not significant
LDH increase	Not significant	Not significant
Infiltrates at the chest X-ray	At onset +	Pneumonia ++

Footnotes: ARDS: acute respiratory distress syndrome, C-RP: C-reactive protein, AST: aspartate aminotransferase, ALT: alanine aminotransferase, LDH: lactate dehydrogenase, ≈: uncertain meaning, +: low risk, ++: average risk, +++: high risk.

**Table 2 jcm-10-01139-t002:** Commercially available diagnostics kits mainly used in Italy in real-time reverse transcription polymerase chain reaction (rRT-PCR) with gene target and limit of detection.

Company (Assay Name)	Gene Target	LoD	Specimen Types	Approval
Abbott Diagnostics (*ID NOW COVID-19*)	RdRp	125 copies/mL	Nasal, throat, NPS	FDA (US)
Abbott Molecular (Abbott RealTime SARS-CoV-2 EUA Test)	RdRp, N	100 virus copies/mL	NPS, OPS, nasal swab, BAL	FDA (US) CE-IVD
Cepheid (*Xpert Xpress SARS CoV-2*)	N2, E	250 copies/mL	NPS, OPS, nasal, mid-turbinate swab, nasal wash/aspirate	FDA (US), Health Canada, Australia, Singapore, Philippines, Brazil
DiaSorin Molecular (LIAISON MDX)	ORF1ab, S gene	NPS: 500 copies/mL, Nasal swab: 242 copies/mL	Nasal swab, NPS, nasal wash/aspirate, BAL	CE-IVD
Tib Molbiol (*Modular DX kit SARS-CoV-2)*	E	1–10 copies/reaction	OPS, NPS	RUO (research use only)
Roche Molecular System (Cobas 6800 SARS-CoV-2)	ORF-1a/b, E	1000 RNA genome equivalents/mL	NPS, OPS	US-FDA, CE-IVD
Seegene (Allplex 2019-nCoV Assay)	RdRp, N, E	100 RNA copies/rxn	NPS, NPA, OPS, sputum, BAL	Korea (Korea CDC), US-FDA, CE-IVD
bioMerieux (ARGENE SARS-CoV-2 R-GENE)	RdRp, N, E	380 genomic copies/mL	NPS	RUO (research use only)

Legend: LoD: limit of detection, RnRp: RNA-dependent RNA polymerase, N: nucleocapsid, E: envelope, ORF: open reading frame, EUA: Emergency Use Authorization, BAL: bronchoalveolar lavage, NPA: nasopharyngeal aspirate, NPS: nasopharyngeal swab, OPS: oropharyngeal swab, FDA: Food and Drug Administration, CE-IVD: European Conformity In-Vitro Diagnostic, CDC: Centers for Disease Prevention and Control.

**Table 3 jcm-10-01139-t003:** COVID-19 disease phase and sample site recommendation.

Sample Sites	Asymptomatic Phase	Onset of the Symptomatic Phase	Symptomatic Phase	Convalescence Phase
Naso-pharyngeal swabs	Unclear	Highly recommended Detection rate: 80%	Recommended Detection rate: 59%	Recommended Detection rate: 36%
Oro-pharyngeal swabs	Unclear	Highly recommended Detection rate: 75%	Not recommended Detection rate: 35%	Not recommended Detection rate: 12%
Saliva collection	Unclear	Highly recommended Detection rate: 82.2%	Unclear	Unclear
Sputum collection	Unclear	Highly recommended Detection rate: 98%	Highly recommended Detection rate: 69%	Not recommended Detection rate: 46%
Bronco-alveolar lavage	Unclear/not recommend	Unclear/not recommended	Highly recommended in intubated patients Detection rate: 94%	Not recommend
Fecal/anal swabs	Not recommend	Not recommended Detection rate: 48%	Not recommended	Recommended Detection rate: 73%

## Data Availability

Not applicable.

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
