# Peer review of "Analysis of the Clinical and Epidemiological Meaning of Screening Test for SARS-CoV-2: Considerations in the Chronic Kidney Disease Patients during the COVID-19 Pandemic"

_jcm, 2021, doi:10.3390/jcm10051139_

Round 1
Reviewer 1 Report
In the current manuscript the authors discuss screening modalities for SARS-CoV-2 in chronic kidney disease patients an interpret the clinical and epidemiological meaning and utility of these tests in this vulnerable population.
this is a well written review and I have a few comments that may improve the quality of this manuscript:
- It is imperative to test for SARS-CoV-2 positivity rates in vulnerable population and one of the major hurdles is a lack of transparency which is often an obstacle to epidemiological analysis. The authors should comment on the reporting rates in these studies and give suggestions about how to improve reporting and thereby establish real prevalence of the infection in the general population as well as specific vulnerable populations.
- In the same context the authors should discuss rapid and increased targeted testing of these vulnerable populations and how this can be done in the setting CKD patients who are more susceptible but at the same time need more medical attention and care despite the pandemic.
- While it is useful to propose high frequency, lower sensitivity testing additional consequences of this approach especially in terms of false positive results in small hospital settings as well as in broad clinical practice must be balanced. Some comment on this would be worthwhile.
- Another important aspect is the pretest probability that patients have the disease. It has been demonstrated that the lower the prevalence of the disease the higher the rate of false positive tests and this in some cases may complicate the analysis and subsequent recommendations for testing. Usually for a population with a given disease prevalence it is straightforward to establish sensitivity and specificity of an assay. However, with the SARS Co V2, it is difficult to establish true prevalence and the prevalence of the disease is changing as well as is different in different populations. These limitations can be acknowledged in the review.
- Just like false negative results, false positive results may also have dire consequences and in the setting of CKD may stress the already stressed patient more. Therefore, confirmatory testing and subsequent investigation of unexpectedly positive results or higher than observed positive results should be done. It would be worthwhile commenting on this.
Author Response
Dear reviewer,
Thank you for your comments and suggestions, which were a stimulus to do better. Especially, we thank you for your advice.
Specifically,
About the prevalence of the disease, we added a paragraph on page 7, lines 318-325. We tried to underline the difficulties in the analysis of prevalence and how to manage them.
About the false positive, we added a paragraph on page 6, lines 232- 339 with some considerations about the consequence of the false-positive rate in the CKD patients and healthcare providers' perception
Finally, About the rapid test for SARS CoV-2, we did not add any notices because this type of test merits in our opinion a paper to describe their real value better.
We hope you can appreciate the added comments and you could find an improvement in the manuscript.
Best regards.
Reviewer 2 Report
Thank you for giving me the opportunity to read and comment a report “Screening test for SARS-CoV-2 in chronic kidney disease patients: an in-depth analysis of clinical and epidemiological meaning”, by Martino et al..
In the reviewed manuscript, the importance of tests for the early diagnosis of COVID-19 is reported, especially in patients with chronic kidney disease. The authors made an in-depth review of the COVID-19 risks in patients with chronic kidney disease and of the screening tests, methods, accuracy, specimen and statistical insights.
This paper is well written, correctly structured and definitely it is of relevance to readers of the journal, in general, and nephrologists in particular.
Congratulations to the authors for this interesting article. Furthermore, some suggested minor changes are included in the comments given below.
- It would be convenient to define some abbreviations from Table 1, such as ARDS, PCR, AST / ALT or LDH
- The resolution of the figures is not good, so it would be advisable to improve it.
- The review made in this article is concrete and concise, but the conclusions are general. Perhaps the authors may give specific recommendations for the nephrologists clinical practice
- It would be advisable to review the bibliography, since, at least, reference 19 is wrong.
Author Response
Dear reviewer,
We want to thank you for your comments and your suggestions, which were detailed and useful.
Specifically, we followed your suggestions:
- We added footnotes in table 1. Furthermore, we revised the other tables.
- We reconfigured figure 1
- We changed the conclusion with specific recommendations for nephrologists
- We edited the bibliography
Thank you again for your punctuality which helps us to improve the manuscript.
Best regards.
Reviewer 3 Report
The authors describe in this review several general aspects of SARS-CoV-2 infection and Covid-19 disease. While the title of the manuscript suggests to the reader that specific details concerning infection and disease in patients with chronic kidney disease will be presented, most parts of the manuscript discuss general topics of the disease, e.g. social impact, symptoms at onset of disease, structure of the virus, and accuracy of testing with rt-PCR. Within these parts of the manuscript, little information on specific details concerning patients with CKD are given.
Except for table 1, where some details are presented that might be specific for CKD patients, no data are presented in this specific population.
In my opinion, this manuscript does not offer relevant new data, and only globally summarizes well known topics. A more detailed description of incidence, symptoms, course of the disease, outcome, and rehabilitation, as well as handling the chronic disease in times of the pandemic would have been interesting for the reader.
Author Response
Dear Reviewer,
Thanks for your revision and your comments, which we read with extreme interest. Consequently, we tried to optimize our manuscript, following your suggestions. Specifically, we changed the title to characterize the document better. We added some paragraph better to describe the SARS CoV-2 infection in CKD patients, and better define symptoms, outcome, and risk factors (as you can see on page 2 lines 50-52, lines 64-68, page 3 lines 119-123, page 6 lines 211-218).
On the other hand, we would like to emphasize that our manuscript is a narrative review, which reports a comprehensive, critical, and objective analysis of the current knowledge on a topic. Specifically, our focus was to analyse the limits of RRT-PCR for SARS CoV-2 as a screening test in CKD patients and raise awareness among nephrologists about the test weakness. Likely these arguments are not surprising but, they merit a deep thought. During COVID-19 pandemic, we have seen too many cases of an uncritical interpretation of negative test as the absence of COVID-19 with consequent increasing of COVID-19 diffusion among patients and health workers. After your comment, we checked how many papers were indexed in PUBMED about "SARS CoV-2 RRT-PCR” and the rates of “Positive predicted value” or “Negative predicted value", and "SARS CoV-2 and accuracy ". We found only few records. Indeed, our manuscript did not discover new things but presented a well-known topic about the screening test accuracy and meaning. However, we hope to induce a reflection on a purely debated aspect as the reliability of SARS CoV-2 screening test., especially in CKD patients.
Thanks again for your comments and suggestions which were a stimulus to improve the level of the manuscript. We hope you can appreciate our effort.
Best regards